# Leaching Behaviors of Calcium and Aluminum from an Ionic Type Rare Earth Ore Using MgSO₄ as Leaching Agent

**Qiang He, Jiang Qiu, Minglu Rao and Yanfei Xiao ***

Faculty of Materials Metallurgy and Chemistry, Jiangxi University of Science and Technology, Ganzhou 341000, China; 6120190242@mail.jxust.edu.cn (Q.H.); 6720180527@mail.jxust.edu.cn (J.Q.); 6720200569@mail.jxust.edu.cn (M.R.)
* Correspondence: xiaoyanfei@jxust.edu.cn

**Abstract:** During the leaching process of ionic rare earth ore (ICREO), ion-exchangeable phase calcium (IEP-Ca) and ion-exchangeable phase aluminum (IEP-Al) are leached along with rare earth, which causes many problems in the enrichment process, such as increasing the precipitant agent consumption and rare earth loss, etc. The agitation leaching kinetics and the column leaching mass transfer process of IEP-Ca and IEP-Al were studied to understand the leaching behavior of impurity in ICREO, which provides guides for the adjustment of the leaching process and to limit the co-leaching of impurities. IEP-Ca and IEP-Al were leached by ion exchange, with the leaching agent cations and the leaching kinetics described by an internal diffusion-controlled shrinking core model with an apparent activation energy of 8.97 kJ/mol and 10.48 kJ/mol, respectively. In addition, a significant reduction in the leaching efficiency of aluminum was caused by the hydrolysis reaction reinforced by the increase in MgSO₄ concentration and temperature. The leaching kinetic data of IEP-Ca and IEP-Al was verified by the column leaching mass transfer process. There was a synchronous increase in the peak concentration of the outflow curve and leaching efficiency of calcium with the concentration of MgSO₄ since IEP-Ca was easily leached. Therefore, as the leaching efficiency of calcium was already very high in the 0.20 mol/L MgSO₄ leaching process, the leaching rate of calcium was limited by the leaching temperature and injection rate of MgSO₄. For aluminum, the hydrolysis of $Al^{3+}$ was promoted by increasing the MgSO₄ concentration and the leaching temperature, thereby effectively reducing the content of aluminum in the leachate.

**Keywords:** ionic-type rare earth ore; calcium; aluminum; leaching behavior; MgSO₄

## 1. Introduction

The ionic-type rare earth ore (ICREO), in which the ion-exchangeable phase rare earth (IEP-RE) accounts for more than 80% of whole-phase rare earth [1], was first discovered in Ganzhou City in 1969. It is mainly located in seven provinces of southern China. In recent years, ICREO has also been found in Laos, Vietnam, Chile, and other countries [2]. ICREO is a valuable strategic mineral resource because of its many advantages: low radioactivity, simple leaching process, complete rare earth partition, and abundance of middle and heavy rare earth elements [3,4]. The development of ICREO can solve the problem that bastnaesite and Baotou mixed rare earth ore can only produce light rare earth and lack medium and heavy rare earth elements [2]. According to the characteristics of the IEP-RE in ICREO being easily desorbed when encountering $Na^+$, $NH_4^+$, and $Mg^{2+}$ and so on [5–7], a series of leaching agents and leaching technologies for ICREO have been put forward by scientists in China. Currently, rare earth is leached from ICREO by an in situ leaching process with ammonium sulfate leaching agent [2]. However, with the increasing requirements of environmental protection, the problem of ammonia nitrogen pollution caused by ammonium sulfate has received increasing attention [8]. A series of strengthening leaching methods and new leaching agents have been developed to reduce or even eliminate the ammonia–nitrogen pollution in the leaching process [6,9–11], among

which the $MgSO_4$ leaching agent, which was proposed by our team [9,12], can maintain the soil nutrients to reduce the dosage of calcium–magnesium fertilizer, and may be able to realize ecologically friendly leaching of ICREO [12,13]. Pilot-plant-scale tests using the $MgSO_4$ leaching agent are being conducted in Yongzhou city (in Hunan Province), Chongzuo city (in Guangxi Province), and Changting city (in Fujian Province).

ICREO also contains ion-exchangeable phase calcium (IEP-Ca) and ion-exchangeable phase aluminum (IEP-Al) [14,15]. Regardless of the type of leaching agent or enhanced leaching technology used, the IEP-Ca and IEP-Al will also be desorbed and exchanged into the leachate in the leaching process of ICREO [16,17], as shown in Equations (1) and (2):

$$\text{Clay·} nCa^{2+}{}_{(s)} + nMg^{2+}{}_{(aq)} \leftrightarrow \text{Clay·} nMg^{2+}{}_{(s)} + nCa^{2+}{}_{(aq)} \tag{1}$$

$$\text{Clay·} 2nAl^{3+}{}_{(s)} + 3nMg^{2+}{}_{(aq)} \leftrightarrow \text{Clay·} 3nMg^{2+}{}_{(s)} + 2nAl^{3+}{}_{(aq)} \tag{2}$$

The concentration of $Ca^{2+}$ and $Al^{3+}$ in the leachate vary with different mines or leaching times. In general, the $Ca^{2+}$ concentration is 0.1–1.0 g/L, and the $Al^{3+}$ concentration is 0.05–2.0 g/L [15,18]. The large amount of impure ions in the leachate will greatly increase the production cost and difficulty of the subsequent enrichment process. For example, if oxalic acid is used as a precipitant to precipitate rare earth in the leachate, $Al^{3+}$ will react with oxalate to form the soluble complex $RE[Al(C_2O_4)_3]$ [16], whereas $Ca^{2+}$ will form $CaC_2O_4$. These will cause an increase in oxalic acid consumption, and a decrease in rare earth yield and decrease in the purity of rare earth concentrate. If carbonate or bicarbonate is used as the precipitant to enrich the rare earth, $Al^{3+}$ and $Ca^{2+}$ can form $Al(OH)_3$ or $CaCO_3$, which will also increase the amount of precipitant and reduce the purity of rare earth products. At the same time, the formation of $Al(OH)_3$ will worsen the crystallization of the precipitate and seriously affect the filtration performance of the precipitate [16]. Moreover, if the solvent extraction method is used to enrich the rare earth in the leachate, it is easy to form colloidal precipitate of aluminum hydroxide in the presence of $Al^{3+}$, which emulsifies the extraction system and makes it impossible to continue the extraction operation [19]. Therefore, impurities, e.g., calcium and aluminum, play a troublesome role in the rare earth processing of ICREO that requires complicated procedures in multiple stages to minimize the content of impurities. Impurity removal from the leachate of ICREO has always been a research hotspot [20].

Many technologies focus on removing impurities from leachate through extraction, precipitation, and other methods, which extend the technological process and reduce the rare earth yield [20]. In addition, more attractive technology has been proposed in recent years to remove impurities by selectively reducing the leaching efficiency of impurities during the leaching process of ICREO. A series of impurity inhibition leaching methods have been developed [2]. However, organic substances, such as acetic acid and sulfosalicylic acid, were commonly used as impurity inhibitors [21,22], leading to the environmental pollution and high cost of the leaching process. To some extent, it is hopeful that rare earth leachate with low impurity content can be obtained utilizing the different leaching behavior of impurity ions and rare earth ions appropriately [18,23]. In the early stage, our research group studied the leaching kinetics, mass transfer characteristics, and leaching process of rare earth with the $MgSO_4$ leaching agent [9,24], but the leaching behavior of calcium and aluminum was not discussed. Therefore, the leaching kinetics of calcium and aluminum in ICREO were determined by agitation leaching experiments in this paper. Moreover, column leaching experiments, used to simulate in situ leaching processes, were employed to study the leaching behaviors of calcium and aluminum, which provided a reference for the in situ leaching process of ICREO and verified the above-mentioned kinetic data. The research in this paper has a certain guiding significance for the separation of rare earth and impure ions in the leaching process.

## 2. Experiment

### 2.1. Characterization of Experimental ICREO

ICREO was collected from a completely weathered layer of the Liutang mine area in Chongzuo City, Guangxi Province. The ore was dried in a drying oven. Generally, a part of rare earth and impurities are present as the ion-exchangeable phase in ICREO [1,4]. To determine the content of IEP-RE and IEP-Ca in the rare earth ore, 300 g ICREO, which was obtained from the dried ore using the quadrate method, was packed in a Φ40 mm column, and leached with 600 mL 40 g/L $(NH_4)_2SO_4$ solution. The effluent was collected and then diluted to 1 L with pure water. The contents of rare earth and calcium in the effluent were analyzed using an ICP-AES. Moreover, 1.0 g ground rare earth ore was placed into a 100 mL centrifuge tube, and subsequently added to 50 mL of a 1.0 mol/L KCl solution. After the centrifuge tube was placed on a Conrad oscillator and shaken for 30 min, the supernatant was obtained by centrifugation. The content of $Al^{3+}$ in the supernatant was measured by ICP-MS to obtain the IEP-Al content in ICREO. The results of the above tests are shown in Table 1 and Figure 1. Table 1 shows that the contents of rare earth, calcium, and aluminum in the ion-exchangeable phase of ICREO are 0.15%, 0.025%, and 0.003%, respectively. Figure 1 shows that the distribution patterns of La, Nd, and Y in the ion-exchangeable phase are high, so the experimental ore is the rare earth ore with middle Y and rich Eu [4]. The phase of the experimental rare earth ore was analyzed by XRD with Cu Kα. The results in Figure 2 show that the main minerals in the ore are quartz, kaolinite, mica, and montmorillonite.

**Table 1.** Main chemical composition of the ion-exchangeable phase in the ICREO sample.

| Element | RE$_2$O$_3$ | Mg | Ca | Al | Fe |
|---|---|---|---|---|---|
| Content (wt. %) | 0.150 | 0.005 | 0.025 | 0.003 | <0.001 |

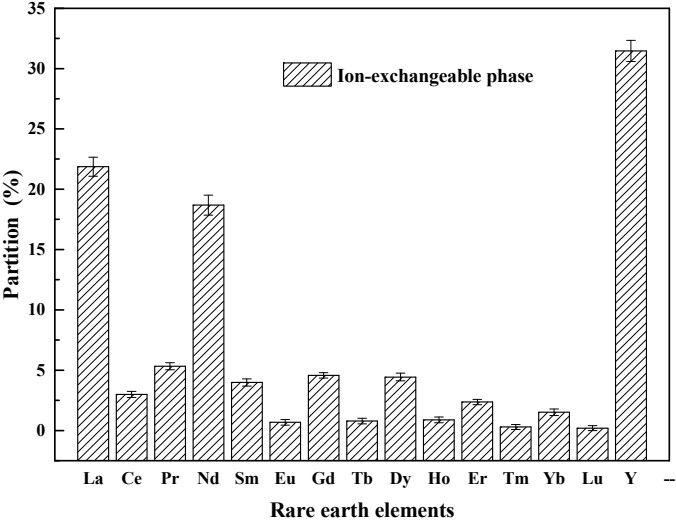

**Figure 1.** Rare earth distribution patterns of the ion-exchangeable phase (mass fraction %).

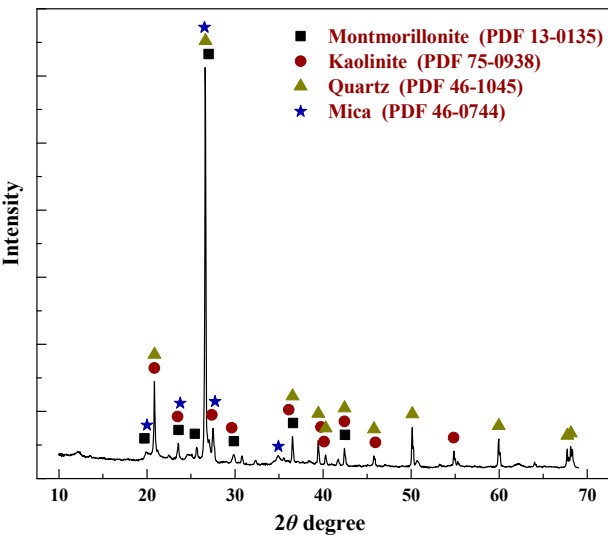

**Figure 2.** XRD pattern of the experimental rare earth ore.

### 2.2. Batch Leaching Experimental Procedure

All the chemicals in the experiments were analytically pure, such as $MgSO_4$ and $HNO_3$, and were purchased from Sinopharm Chemical Reagent Co., Ltd, Shanghai, China.

In the agitation leaching experiments, the ICREO was simply ground to obtain an average particle size of 0.6–0.9 mm. A 500 mL three-necked flask was used as the reactor. It was fitted with a thermometer and an agitator to monitor temperature and adjust stirring speed. A constant temperature water bath (DCW-4006, Shanghai Bilon Instrument Manufacturing Co., Ltd., Shanghai, China) was applied to control the leaching temperature at 10–55 °C. At the beginning of the experiment, 200 mL 0.20 mol/L $MgSO_4$ leaching agent was placed into the flask and preheated to the desired temperature by the water bath. The stirring speed was controlled in a certain range by the agitator. Then, 40 g treated ICREO was rapidly placed in the reactor, and the leaching time was immediately recorded. After a certain reaction time, a 5 mL sample was taken from the reactor and quickly added to a 100 mL centrifuge tube containing 50 mL pure water. Finally, a solid–liquid separation of the sample was conducted by centrifugation (5804(R)/5810(R), Eppendorf). The supernatant fluid was diluted to 100 mL and the concentrations of calcium and aluminum in the supernatant fluid were tested by ICP-AES [25].

In the column leaching experiments, 40 mm-inner-diameter columns with the heat preservation tube were applied as the leaching equipment, and precision pumps were applied to control the flow rate of the leaching agent. An $MgSO_4$ solution with a given concentration ($C_0$) was used as the leaching agent. At the beginning of the experiment, 300 g dried ICREO was packed into the column, and the column was slightly tapped to make sure that the packed bed heights were the same in each experiment. Then, the column with ICREO was eluted with the above $MgSO_4$ solution at a specified temperature and a desired leaching flow rate ($v$). Finally, every 25 mL leachate was collected from the bottom of the column as one sample, and the concentration of $Ca^{2+}$ and $Al^{3+}$ in the samples were tested by ICP-AES [25].

From the tests, the leaching efficiency of impurities (calcium and aluminum) can be defined as:

$$\eta = \varepsilon_V/\varepsilon_0 \text{ or } \eta = \varepsilon_t/\varepsilon_0 \tag{3}$$

where $\eta$ (%) is the leaching efficiency of calcium and aluminum, $\varepsilon_0$ (g) is the amount of calcium and aluminum that exist in the ion-exchangeable phase in the experimental ore, $\varepsilon_V$ (g) is the total amount of calcium and aluminum in the leachate when the collected volume is V (mL) in the column leaching experiments, and $\varepsilon_t$ (g) is the total amount of calcium and aluminum in the leachate when the leaching time is $t$ (s) in the agitation leaching experiments.

## 3. Results and Discussion

### 3.1. Theoretical Analysis of Precipitation Behaviors of Calcium and Aluminum

In the leaching process of ICREO, the IEP-Ca and IEP-Al are desorbed by $MgSO_4$ and enter the leachate [16]. Because there are a lot of sulfate ions, the $Ca^{2+}$ in the leachate may form calcium sulfate precipitate. Moreover, $Ca^{2+}$ and $Al^{3+}$ will hydrolyze to form hydroxide precipitate in a solution of a certain alkalinity. Therefore, to clearly understand the behaviors of impurities in the leaching process of ICREO, the theoretical analysis of precipitation of impurities in the solution was developed first.

### 3.1.1. Precipitation Behavior of Calcium

$Ca^{2+}$ is a kind of strong alkaline ion, which generally does not hydrolyze in water. It exists in the form of simple $Ca^{2+}$ in solution. Based on the solubility product of calcium hydroxide at different temperatures, the pH-$lgC_{Ca}$ diagram of $Ca^{2+}$ was plotted and is presented in Figure 3. It indicates that calcium hydroxide is formed if the pH value is greater than 11 and the concentration of $Ca^{2+}$ is 1 mol/L in the solution. However, the concentration of $Ca^{2+}$ in leachate hardly exceeded 0.1 mol/L and the pH was less than 6.0 during the leaching process by $MgSO_4$, which led to little possibility of hydroxide precipitation generating. However, calcium sulfate precipitate may generate with a concentration of sulfate ions of about 0.2 mol/L in the leachate. For this purpose, the solubility of calcium sulfate in the $CaSO_4$-$MgSO_4$-$H_2O$ ternary phase system was measured previously [26]. The solubility of calcium sulfate was sharply decreased with an increase in the concentration of $MgSO_4$ at different temperatures. The minimal solubility, in the range of 0.009–0.012 mol/L, was obtained with the concentration of $MgSO_4$ fluctuating from 0.06 mol/L to 0.11 mol/L. Thence, the calcium sulfate precipitate formed above the threshold concentration (360 mg/L) of $Ca^{2+}$.

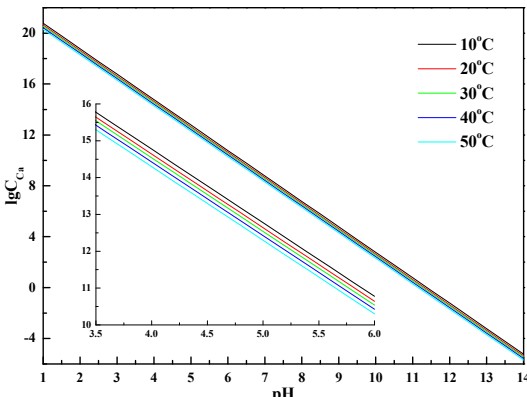

**Figure 3.** pH-$lgC_{Ca}$ diagram of $Ca^{2+}$ in solution at different temperatures.

### 3.1.2. Precipitation Behavior of Aluminum

The solubility of total aluminum in solution is determined by dissolution reaction of amorphous aluminum hydroxide $Al(OH)_3$ (am) via Equations (4) and (5), whose equilibrium is a function of various aluminum species. A brief description of the plotting steps for the solubility diagram of $Al^{3+}$ is summarized in this article, and all of the mentioned data were published previously [27]. A set of the most common hydroxyl aluminum species, including $Al_3(OH)_4^{5+}$, $Al_2(OH)_2^{4+}$, $Al(OH)^{2+}$, $Al(OH)_2^+$, $Al(OH)_3^0(aq)$, and $Al(OH)_4^-$, was selected to be calculated, with the more complex multinuclear compounds of aluminum merely formed by the polymerization and flocculation of these simple species with the changing pH value in the solution. Therefore, the total amount of aluminum dissolved in the solution is the sum of $Al^{3+}$ and all the above hydroxylated aluminum species (Equation (6)). The chemical equilibrium equations of the stepwise hydrolysis of $Al^{3+}$ are given in Equations (7)–(12), and the logarithmic concentration of aluminum species $lgC_{Al_m(OH)_n^{(3m-n)+}}$ was calculated by Equation (13). The detailed data related to the

dissolution reaction of the amorphous aluminum hydroxide and hydrolysis of $Al^{3+}$ are listed in Table 2 [27]. Based on these data, the solubility diagram of aluminum was built and is illustrated in Figure 4. The concentration of total aluminum in the solution showed a decreasing trend first and was followed by an increase as the pH value increased from 0 to 14, and a lower concentration of about 2.75 μmol/L (0.074 mg/L) was obtained at pH 5.70. Therefore, the precipitate may be generally formed from the hydrolysis of $Al^{3+}$ in leachate.

$$Al(OH)_3 \ (am) = Al^{3+} + 3OH^-, \ K_{sp} \tag{4}$$

$$lgC_{Al^{3+}} = lgK_{sp} - 3\,lgK_w - 3\,pH \tag{5}$$

$$\textstyle\sum Al\ (aq) = [Al_3(OH)_4{}^{5+}] + [Al_2(OH)_2{}^{4+}] + \left[Al^{3+}\right] + \left[Al(OH)^{2+}\right] + [Al(OH)_2{}^+] + \\ [Al(OH)_3{}^0] + [Al(OH)_4{}^-] \tag{6}$$

$$3Al^{3+} + 4H_2O = Al_3(OH)_4{}^{5+} + 4H^+, \ K_{3\text{-}4} \tag{7}$$

$$2Al^{3+} + 2H_2O = Al_2(OH)_2{}^{4+} + 2H^+, \ K_{2\text{-}2} \tag{8}$$

$$Al^{3+} + H_2O = Al(OH)^{2+} + H^+, \ K_{1\text{-}1} \tag{9}$$

$$Al^{3+} + 2H_2O = Al(OH)_2{}^+ + 2H^+, \ K_{1\text{-}2} \tag{10}$$

$$Al^{3+} + 3H_2O = Al(OH)_3{}^0(aq) + 3H^+, \ K_{1\text{-}3} \tag{11}$$

$$Al^{3+} + 4H_2O = Al(OH)_4{}^- + 4H^+, \ K_{1\text{-}4} \tag{12}$$

$$lgC_{Al_m(OH)_n^{(3m-n)+}} = lgK_{m\text{-}n} + m\,lgK_{sp} - 3m\,lgK_w + (n - 3m)\,pH \tag{13}$$

**Table 2.** The parameters and equations of the stepwise hydrolysis of $Al^{3+}$.

| Species | Equilibrium Constants | | Equations |
| --- | --- | --- | --- |
| | lgK | 25 °C | 25 °C |
| $Al_3(OH)_4^{5+}$ | $lgK_{3\text{-}4}$ | −13.9 | $lgC_{Al(OH)_4^{5+}} = 17.81 - 5\,pH$ |
| $Al_2(OH)_2^{4+}$ | $lgK_{2\text{-}2}$ | −7.1 | $lgC_{Al(OH)_2^{4+}} = 14.04 - 4\,pH$ |
| $Al^{3+}$ | $lgK_{sp}$ | −31.43 | $lgC_{Al^{3+}} = 10.57 - 3\,pH$ |
| $Al(OH)^{2+}$ | $lgK_{1\text{-}1}$ | −4.96 | $lgC_{Al(OH)^{2+}} = 5.61 - 2\,pH$ |
| $Al(OH)_2^+$ | $lgK_{1\text{-}2}$ | −10.91 | $lgC_{Al(OH)_2^+} = -0.34 - pH$ |
| $Al(OH)_3^0(aq)$ | $lgK_{1\text{-}3}$ | −17.02 | $lgC_{Al(OH)_3^0(aq)} = -6.45$ |
| $Al(OH)_4^-$ | $lgK_{1\text{-}4}$ | −22.83 | $lgC_{Al(OH)_4^-} = pH - 12.26$ |

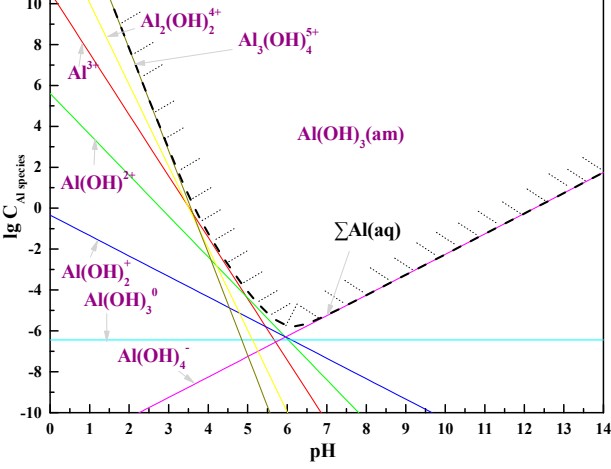

**Figure 4.** Theoretical aluminum solubility diagram in equilibrium with amorphous aluminum hydroxide $Al(OH)_3$ (am), 25 °C.

### 3.2. Leaching Kinetics of Calcium and Aluminum

In ICREO, some non-rare earth ions adsorbed on clay minerals in the form of an ion-exchangeable phase are easily desorbed by other cations [28]. The chemical reaction equation is shown as Equations (1) and (2). Therefore, the leaching process of calcium and aluminum in ICREO is mainly the ion exchange or desorption process of absorbed $Ca^{2+}$ (or $Al^{3+}$) caused by $Mg^{2+}$ in the leaching agent. It is assumed that the mineral particles of ICREO formed by the loose combination of various clay particles are approximately spherical. A shrinking-core model is usually used to describe the leaching process [24,29]. The leaching process is divided into five stages: the outer diffusion of $Mg^{2+}$ through liquor film, the inner diffusion of $Mg^{2+}$ inside mineral particles, the chemical reaction of ion exchange on the particle surface, the inner diffusion of exchanged calcium and aluminum inside mineral particles, and the outer diffusion of calcium and aluminum through liquor film to leachate. The leaching rate of calcium (or aluminum) with $MgSO_4$ leaching agent may be controlled by chemical reaction, outer diffusion, or inner diffusion, and the corresponding rate control equation is expressed in Equations (14)–(16).

Moreover, temperature strongly affects the reaction rate constant, shown as the Arrhenius equation (Equation (17)). According to the Arrhenius law, the rate control step can be verified by calculating the apparent activation energy. When the apparent activation energy is 4–12 kJ/mol, the leaching rate may be controlled by the diffusion steps, and when it is more than 40 kJ/mol, the leaching rate may be controlled by the chemical reaction [30]. Therefore, a series of experiments at different temperatures and stirring speeds was performed to study the leaching kinetics of calcium and aluminum.

$$(a)\ \text{Chemical reaction controls}:\ 1 - (1 - \eta)^{1/3} = k_1 t \tag{14}$$

$$(b)\ \text{Outer diffusion controls}:\ \eta = k_2 t \tag{15}$$

$$(c)\ \text{Inner diffusion controls}:\ 1 - (2/3)\eta - (1 - \eta)^{2/3} = k_3 t \tag{16}$$

$$k = A\,e^{-\frac{E}{RT}}\ \text{or}\ \ln k = \ln A - \frac{E}{RT} \tag{17}$$

where $k_1$, $k_2$, and $k_3$ (s$^{-1}$) are the apparent leaching reaction rate constants for different control steps; E is the apparent activation energy (kJ/mol); $T$ is the leaching temperature (K); R is the ideal gas constant (J/(mol·K)); and $A$ is the apparent pre-exponential factor.

#### 3.2.1. Leaching Kinetics of Calcium

The leaching process of calcium only involves the desorption of IEP-Ca [13]. To investigate the leaching kinetics of calcium, the effect of temperature on the leaching efficiency of calcium was examined. In the experiments, the stirring speed was 400 r/min and the leaching temperature was in the range of 10–55 °C. The leaching efficiency curve of calcium at different temperatures is shown in Figure 5. As shown in Figure 5, at any temperature, the leaching efficiency of calcium rapidly increased in the first 0–200 s; then, it slowly increased to more than 95% and tended to balance. A higher leaching temperature leads to a higher leaching rate (the tangent of the leaching efficiency and time curve at time t), so a leaching efficiency of 95% requires less time.

The leaching data of calcium were entered into the kinetic equations of different control models by the trial-and-error method. The behavior of $1 - (2/3)\eta - (1 - \eta)^{2/3} = k_3 t$ with the leaching time was linear, as shown in Figure 6A. The linear correlation coefficient ($R^2$) was greater than 0.99, which implies that the data of the leaching efficiency fit well with Equation (17). Thus, the leaching rate was controlled by the inner diffusion. Moreover, Figure 6A shows that the k increased with the increase in leaching temperature, whose values were $9.93 \times 10^{-4}$ (10 °C), $1.18 \times 10^{-3}$ (25 °C), $1.34 \times 10^{-3}$ (35 °C), $1.50 \times 10^{-3}$ (45 °C), and $1.67 \times 10^{-3}$ (55 °C). Moreover, according to Equation (17), the apparent activation energy of the leaching process can be achieved by plotting $\ln k_3$ vs. $1/T$ for the five temperatures. The slope of the straight line shown in Figure 6B was 1.079, so the apparent activation

energy was 8.97 kJ/mol, which is within the conventional activation energy of the inner diffusion-controlled leaching processes (4–12 kJ/mol) [30]. Compared with the previous leaching kinetics data of rare earth by MgSO$_4$ under the identical conditions, the apparent activation energy of calcium was slightly lower than that of rare earth (9.48 kJ/mol) and the apparent leaching rate constant of calcium was larger at all temperature [24]. In addition, the time required for the leaching efficiency of calcium to reach equilibrium was shorter than that of rare earth [24]. The main reason is that the Ca$^{2+}$ is divalent with a small charge density, and the adsorption ability of clay minerals to Ca$^{2+}$ is weak; therefore, IEP-Ca in ICREO is more easily desorbed than rare earth and IEP-Al by MgSO$_4$ [31].

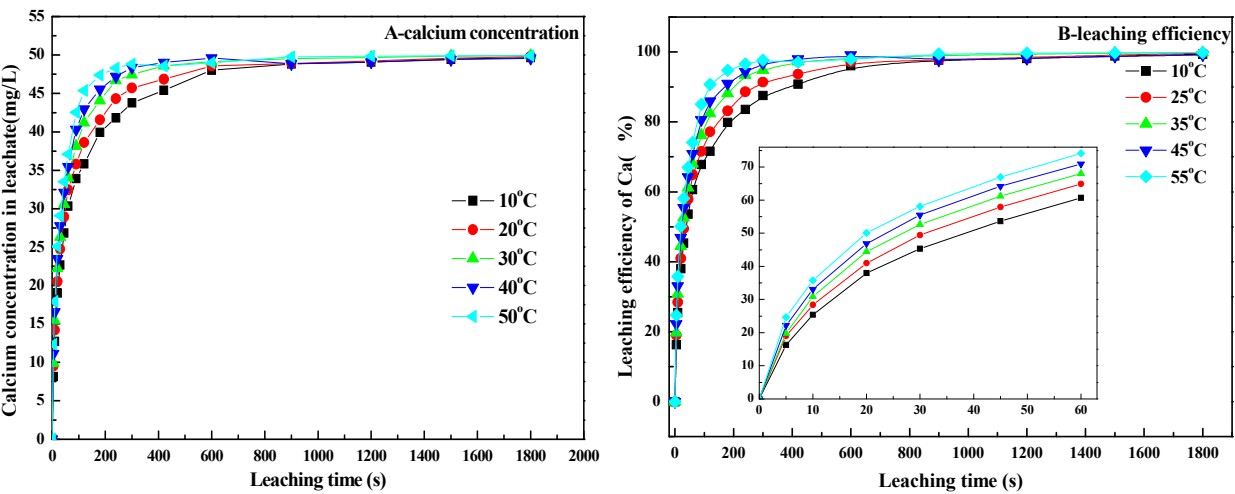

**Figure 5.** Effect of temperature on the leaching of calcium (C$_0$ = 0.20 mol/L, stirring speeds 400 r/min).

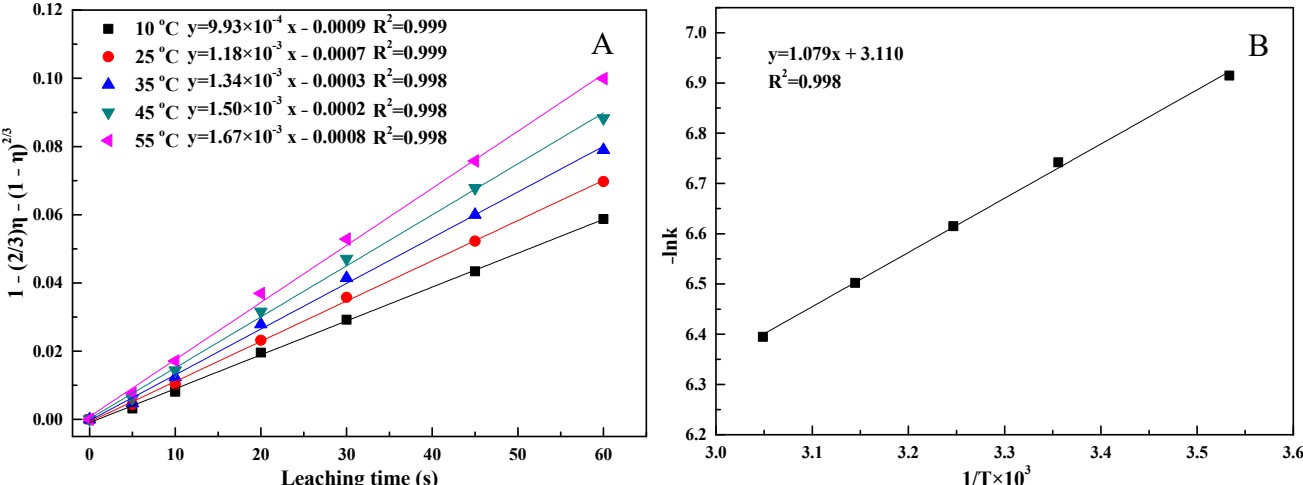

**Figure 6.** Leaching kinetic analysis of calcium at different temperatures. (**A**) plots of $1 - (2/3)\eta - (1 - \eta)^{2/3}$ vs. time; (**B**) Arrhenius plot.

In addition, the effect of the stirring speed on the leaching efficiency of calcium was studied at 25 °C. The results Figure 7 shows that the leaching efficiency curve of calcium basically coincided at different stirring speeds, which implies that the stirring speed has no effect on the leaching of calcium. This phenomenon further determined that the leaching of calcium is controlled by inner diffusion. Compared with RE$^{3+}$, clay minerals have a weak adsorption capacity for Ca$^{2+}$ due to their divalent property [31], so the leaching rate of calcium is faster. Therefore, Ca$^{2+}$ is inevitably leached in the process of rare earth leaching.

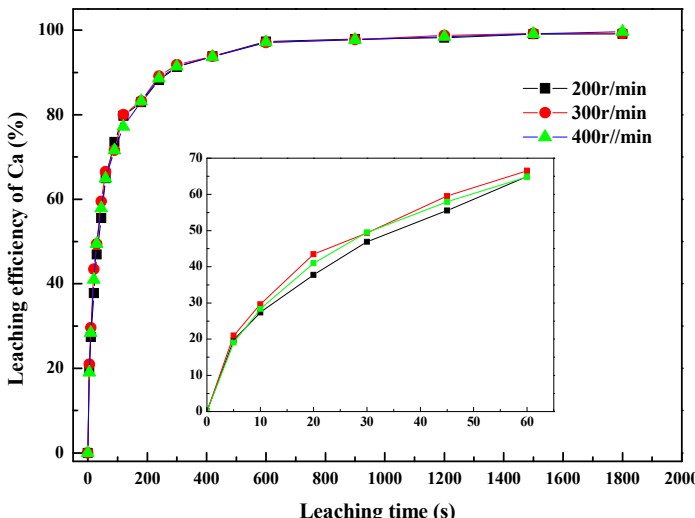

**Figure 7.** Effect of the stirring speeds on the leaching efficiency of calcium ($C_0 = 0.20$ mol/L, T = 25 °C, stirring speeds 400 r/min).

### 3.2.2. Leaching Kinetics of Aluminum

The effect of temperature on the leaching efficiency of aluminum is presented in Figure 8. The leaching curve of aluminum is quite different from that of calcium. As shown in Figure 8, when the leaching temperature was below 25 °C, the leaching efficiency of aluminum rapidly increased to approximately 40% in the first 60 s, then slowly increased to over 55% with the increase in time, and finally tended to balance. It should be noted that the leaching efficiency at 10 °C was lower than that at 20 °C in the initial stage; however, the leaching efficiency at 10 °C was approximately 5% higher than that at 20 °C in the later stage. When the leaching temperature was higher than or equal to 35 °C, the leaching efficiency of aluminum rapidly increased to approximately 35% in the first 40 s. At this stage, a higher temperature led to a higher leaching rate and a higher leaching efficiency. However, as the leaching time increased, the leaching efficiency first slowly increased and then subsequently decreased. When the leaching time was 1800 s, a higher leaching temperature corresponded to a lower the leaching efficiency of aluminum. For example, the leaching efficiency was 48.8%, 49.8%, and 54.3% at 55 °C, 45 °C, and 35 °C, respectively.

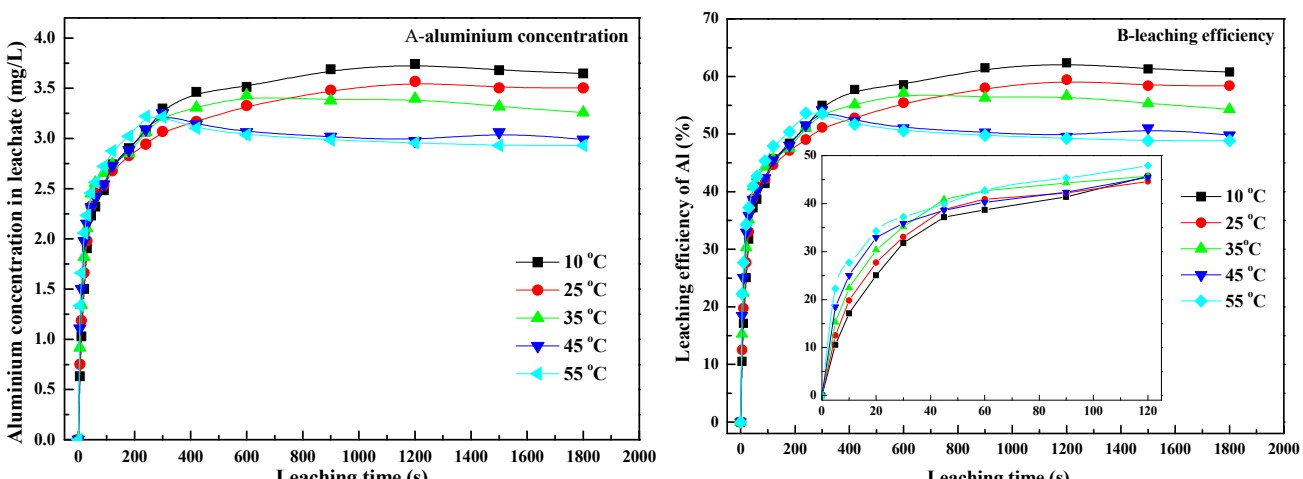

**Figure 8.** Effect of temperature on the leaching of aluminum ($C_0 = 0.20$ mol/L, stirring speeds 400 r/min).

In the $MgSO_4$ leaching system, the nature pH of leachate is less than 6.0 without a pH adjustment. Under these circumstances, the leaching of aluminum contains two reactions: ion exchange and hydrolysis [24], which have a competitive relationship. The

hydrolysis of $Al^{3+}$ is a very complex process that is closely related to the reaction time, pH, concentration of $Al^{3+}$, and leaching temperature [32,33]. According to the results in Figure 4, when the concentration of $Al^{3+}$ in the solution was 5 mg/L, the pH of the initial hydrolysis precipitation of $Al^{3+}$ was about 4.80 at 25 °C. A higher concentration of $Al^{3+}$ corresponded to a smaller initial precipitation pH. In the initial stage of the leaching process, the concentration of $Al^{3+}$ in the system was not sufficiently high and the ion-exchange reaction was dominant. Aluminum mainly existed in the form of stable $Al^{3+}$, $Al(OH)^{2+}$, $Al(OH)_2^+$, etc., in the liquor. With the increase in the leaching time, the concentration of $Al^{3+}$ in the leachate increased, so the leaching efficiency of aluminum rapidly increased. However, with the increase in $Al^{3+}$ concentration, the trend of hydrolysis reaction was enhanced; more $Al^{3+}$ participated in the hydrolysis reaction and formed aluminum hydroxide precipitate, leading to a decrease in the concentration of total aluminum in the leachate. Therefore, as the leaching time increases, the leaching efficiency of aluminum first increases and then decreases, especially when the leaching temperature is higher than 35 °C. Moreover, due to the hydrolysis of aluminum, the leaching efficiency of aluminum is less than 60% at any temperature. Since the hydrolysis of $Al^{3+}$ is endothermic, a higher leaching temperature makes $Al^{3+}$ or $Al(OH)_n^{(3-n)-}$ more easier to collide and agglomerate, and causes a more obvious tendency of hydrolysis. Finally, as shown in Figure 8, with the increase in leaching temperature, the leaching efficiency of aluminum decreased, which was caused by the hydrolysis of $Al^{3+}$. The aforementioned phenomenon conforms with the results of Xu [34].

In the leaching process, only IEP-Al (the content of water-soluble phase aluminum is very low) participates [4]. Since the leaching efficiency of aluminum at the initial stage of the leaching process is only affected by the ion-exchange reaction, the leaching kinetics of aluminum at this stage should also be controlled by inner diffusion, similar to rare earth and calcium. To verify that the kinetics of aluminum leaching is also an inner diffusion-controlled process, the leaching efficiency of aluminum at different temperatures was analyzed. Figure 9A shows plots of $1 - (2/3)\eta - (1 - \eta)^{2/3}$ vs. time for different temperatures of aluminum. The behavior of $1 - (2/3)\eta - (1 - \eta)^{2/3}$ with the leaching time was linear, which indicates that the leaching rate of aluminum is also controlled by inner diffusion. It can be seen from the slope of the line that the apparent reaction rate constants at different temperatures were $4.30 \times 10^{-4}$ (10 °C), $14.64 \times 10^{-4}$ (25 °C), $5.55 \times 10^{-4}$ (35 °C), $7.08 \times 10^{-4}$ (45 °C), and $7.51 \times 10^{-4}$ (55 °C). According to Arrhenius equation, the activation energy of the ion-exchange reaction of $Al^{3+}$ was 10.48 kJ/mol, which is presented in Figure 9B. Due to the higher valence state of hydrated $Al^{3+}$, the charge density of hydrated $Al^{3+}$ is larger than that of calcium, so clay minerals have a greater adsorption ability on $Al^{3+}$ [5,31]. Therefore, the aluminum leaching process has a larger activation energy and a smaller leaching rate. However, the $R^2$ values in the plots of $1 - (2/3)\eta - (1 - \eta)^{2/3}$ vs. time (especially the experimental data of 55 °C) and $\ln k_3$ vs. $1/T$ were slightly small, implying that the fit of the correlation line is relatively low, which may have been caused by the hydrolysis reaction of $Al^{3+}$. Similarly, the effect of the stirring speed on the leaching efficiency of aluminum was conducted at 25 °C, and the results are shown in Figure 10. It was found that the stirring speed had little effect on the leaching of aluminum, which further proves that the leaching of aluminum in the early stage is controlled by inner diffusion.

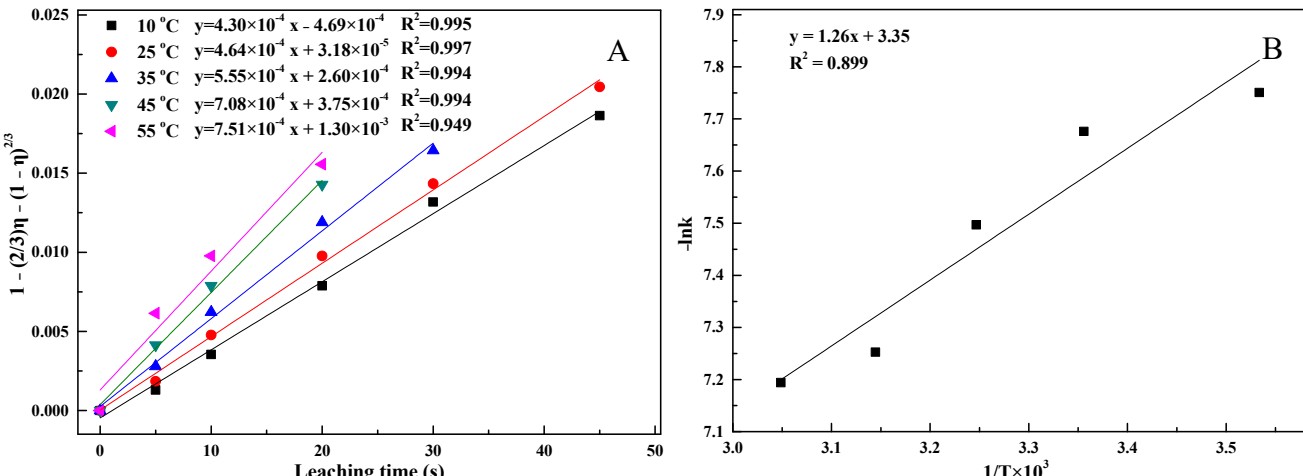

**Figure 9.** Leaching kinetics analysis of aluminum under different temperatures (**A**: plots of $1 - (2/3)\eta - (1 - \eta)^{2/3}$ vs. time; **B**: Arrhenius plot).

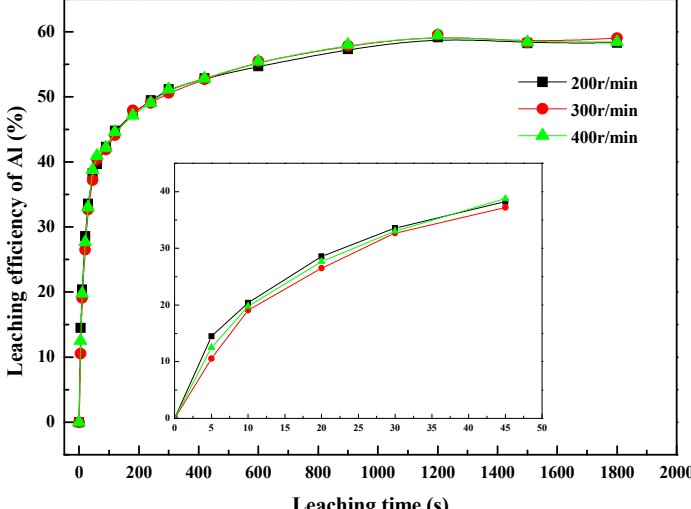

**Figure 10.** Effect of stirring speeds on the leaching efficiency of aluminum ($C_0$ = 0.20 mol/L, T = 25 °C, stirring speeds 400 r/min).

Therefore, the above data show that the tendency of the hydrolysis reaction can be increased by increasing the leaching temperature. In this way, the soluble aluminum content in the leachate and the leaching efficiency of aluminum during the leaching process will reduce.

### 3.3. Leaching Behaviors of Calcium and Aluminum in Columns

Nowadays, the in situ leaching process is applied to recover rare earth from ICREO in industry [8]. To simulate the in situ leaching process, column leaching was used in this paper. The effects of the $MgSO_4$ concentration, flow rate, and leaching temperature on the elution of calcium and aluminum were investigated to verify the above kinetic data and provide a reference for the actual leaching process.

#### 3.3.1. Effect of the $MgSO_4$ Concentration on the Leaching Behaviors of Calcium and Aluminum

Concentration is a key factor that affects the exchangeability and diffusivity of the leaching agent on the surface of mineral particles [18]. The effect of the $MgSO_4$ concentration on the leaching behaviors of calcium and aluminum was determined and the results are shown in Figure 11.

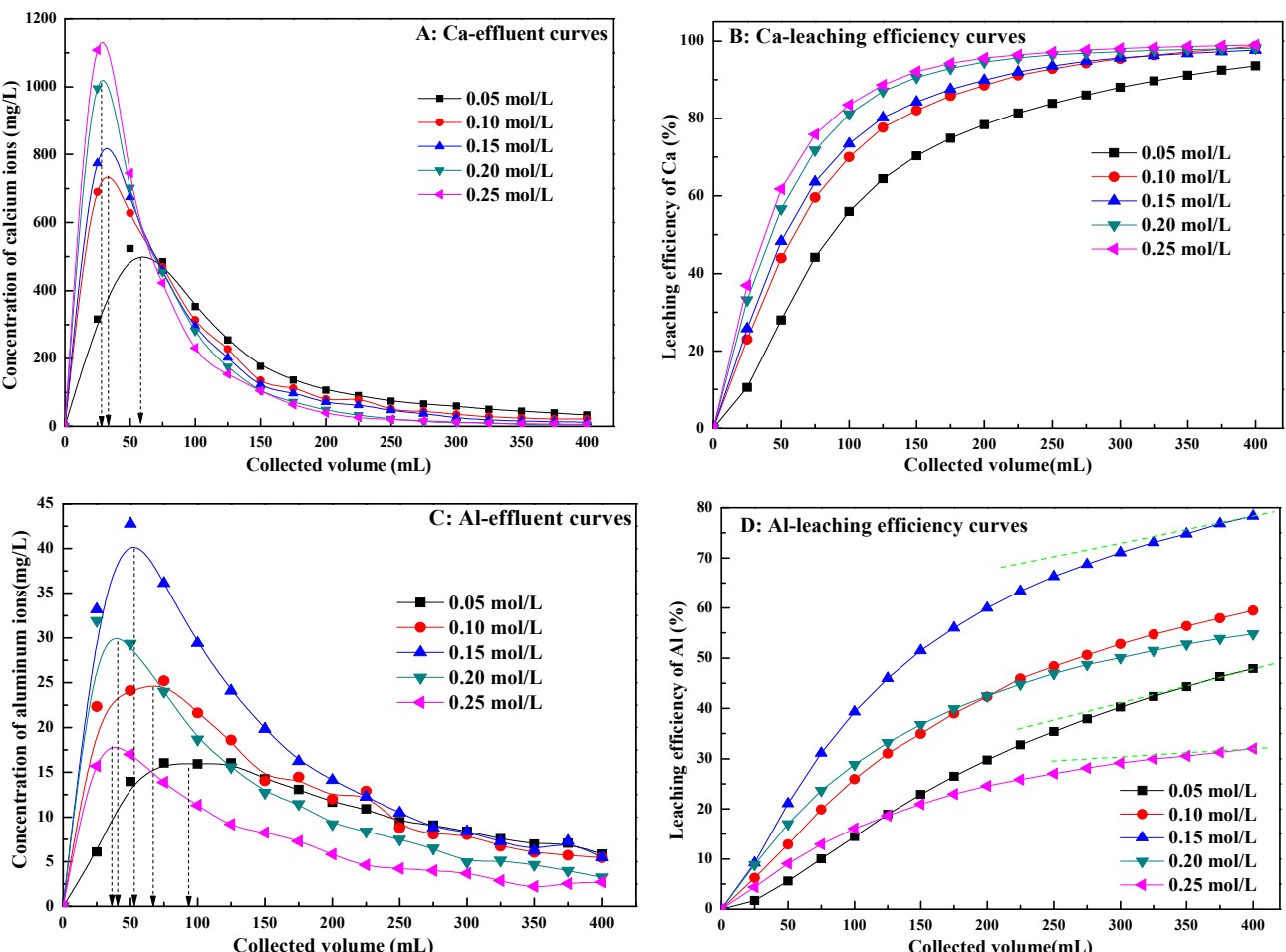

**Figure 11.** Column leaching behaviors of calcium and aluminum with different MgSO$_4$ concentrations (T = 20 °C, υ = 0.60 mL/min, pH = 5.70). (**A**) Ca-effluent curves, (**B**) Ca-leaching efficiency curves, (**C**) Al-effluent curves, (**D**) Al-leaching efficiency curves).

As shown in Figure 11A, the peak concentration of Ca$^{2+}$ increased with the concentration of MgSO$_4$, whereas the volume-to-peak decreased. For example, in response to an increase in MgSO$_4$ concentration from 0.1 mol/L to 0.2 mol/L, the peak concentration and volume-to-peak of Ca$^{2+}$ changed from about 750 mg/L and 28 mL to approximately 1000 mg/L and 26 mL, respectively. Since the leaching process of calcium is controlled by inner diffusion, the concentration gradient on both sides of the product layer was increased with the MgSO$_4$ concentration, thereby reducing the diffusion resistance. Moreover, increasing reactant concentration is beneficial to the forward reaction of chemical equilibrium, so the equilibrium of Equation (1) shifted to the right with the increase in the MgSO$_4$ concentration, and more IEP-Ca was leached. As shown in Figure 11B, the leaching rate (the slope of the leaching efficiency curve) and leaching efficiency of calcium increased with the concentration of MgSO$_4$ under the same collected volume. When the MgSO$_4$ concentration was 0.20 mol/L, the leaching efficiency of calcium exceeded 98% at the collected volume of 250 mL.

For aluminum, its leaching process involves the ion-exchange and hydrolysis reaction. When the concentration of MgSO$_4$ was below 0.15 mol/L, the content of Al$^{3+}$ in the leachate was so low that aluminum hydroxide remained unsaturated, resulting in an inconspicuous hydrolysis reaction of aluminum ions. Therefore, temperature-affected ion-exchange reactions were dominant at this stage. The leaching rate of Al$^{3+}$ accelerated by increasing the concentration of MgSO$_4$, which resulted in an increase in the peak concentration of Al$^{3+}$ and the shift left of volume to peak in Figure 11C. However, when

the $MgSO_4$ concentration was higher than 0.15 mol/L, increasing $MgSO_4$ concentration resulted in an earlier and lower peak concentration of $Al^{3+}$, i.e., the peak concentration of $Al^{3+}$ dropped from 40 mg/L to 30 mg/L as the concentration of $MgSO_4$ increased to 0.20 mol/L. This situation is not surprising. The concentration of $Al^{3+}$ in the liquor was rapidly increased by the accelerated ion exchange reaction during the leaching process; meanwhile, the pH of the leachate was generally between 4.5 and 5.0 [9]. In this case, the supersaturated state of aluminum hydroxide was destroyed and the hydrolysis reaction was greatly promoted. The solubility of total aluminum was lower than $10^{-3}$ mol/L at the pH value in the range of 4.5 to 5.0 (Figure 4), indicating the formation of amorphous aluminum hydroxide precipitate. As a result, the aluminum concentration in the leachate decreased by increasing the concentration of $MgSO_4$.

In Figure 11D, when the $MgSO_4$ concentration was 0.15 mol/L and the collected volume of the leachate was 400 mL, the leaching efficiency of aluminum was less than 80%. However, the leaching efficiency decreased to only approximately 30% at an $MgSO_4$ concentration of 0.25 mol/L. Moreover, Figure 11D suggests that the leaching efficiency of aluminum still tended to increase when the collected volume increased beyond 400 mL. Thus, when the collected volume was 400 mL, a certain amount of IEP-Al remained in the ore, which could be desorbed and enter the liquor, so the leaching rate of $Al^{3+}$ was slower than that of calcium. Furthermore, with the increase in the $MgSO_4$ concentration, the increasing trend of the leaching efficiency of aluminum was weakened. To some extent, it was also illustrated that with the increase in the $MgSO_4$ concentration, more IEP-Al was desorbed. However, the desorbed aluminum just partly entered the leachate, and the other part was hydrolyzed into aluminum hydroxide and remained in the tailings, which led to a lower leaching efficiency.

According to the previous study, the rare earth leaching efficiency exceeded 92% at an MgSO4 concentration of 0.20 mol/L, and then increasing the MgSO4 concentration hardly affected the leaching efficiency of rare earth [9,24]. Therefore, a large quantity of IEP-Ca inevitably entered the leachate in the rare earth leaching. For aluminum, both the leaching rate and leaching efficiency increased with the concentration of $MgSO_4$, which increased the content of total aluminum in the leachate. Thus, the hydrolytic tendency of $Al^{3+}$ was improved, and the concentration of aluminum in leachate was reduced.

### 3.3.2. Effect of the Flow Rate of the Leaching Agent on the Leaching Behaviors of Calcium and Aluminum

In the in situ leaching process, the flow rate of the leaching agent is adjusted according to the specific leaching situation. To simulate the actual process, the effect of the flow rate of the leaching agent on the leaching behaviors of calcium and aluminum was studied, and the effluent curves are presented in Figure 12. In the leaching process, the contact time of the leaching agent with ICREO and the flow direction may affect leaching efficiency significantly. When the flow rate is small, the leaching agent can be in full contact with ICREO, but it is easy for a readsorption phenomenon to occur [35]. Therefore, as shown in Figure 12, due to comprehensive factors, the flow rate hardly affected the leaching behaviors of calcium and aluminum. For $Ca^{2+}$, the peak concentration of $Ca^{2+}$ slightly increased with the decrease in flow rate, as shown in Figure 12A. The ion exchange desorption of ICREO hosted ions was presumably more predominant in the leaching process than other factors such as readsorption. The leaching efficiency curve in Figure 12B shows that the leaching efficiency of calcium slightly decreased with the increase in flow rate, and the collected volume of leachate required for the leaching efficiency of calcium to reach 98% increased.

For $Al^{3+}$, solid–liquid contact time was increased by reducing the flow rate, so it was beneficial to the ion exchange of $Al^{3+}$. However, due to the comprehensive effect of ion exchange and hydrolysis, the flow rate of the leaching agent had little effect on the leaching behavior of $Al^{3+}$, as shown in Figure 12C,D.

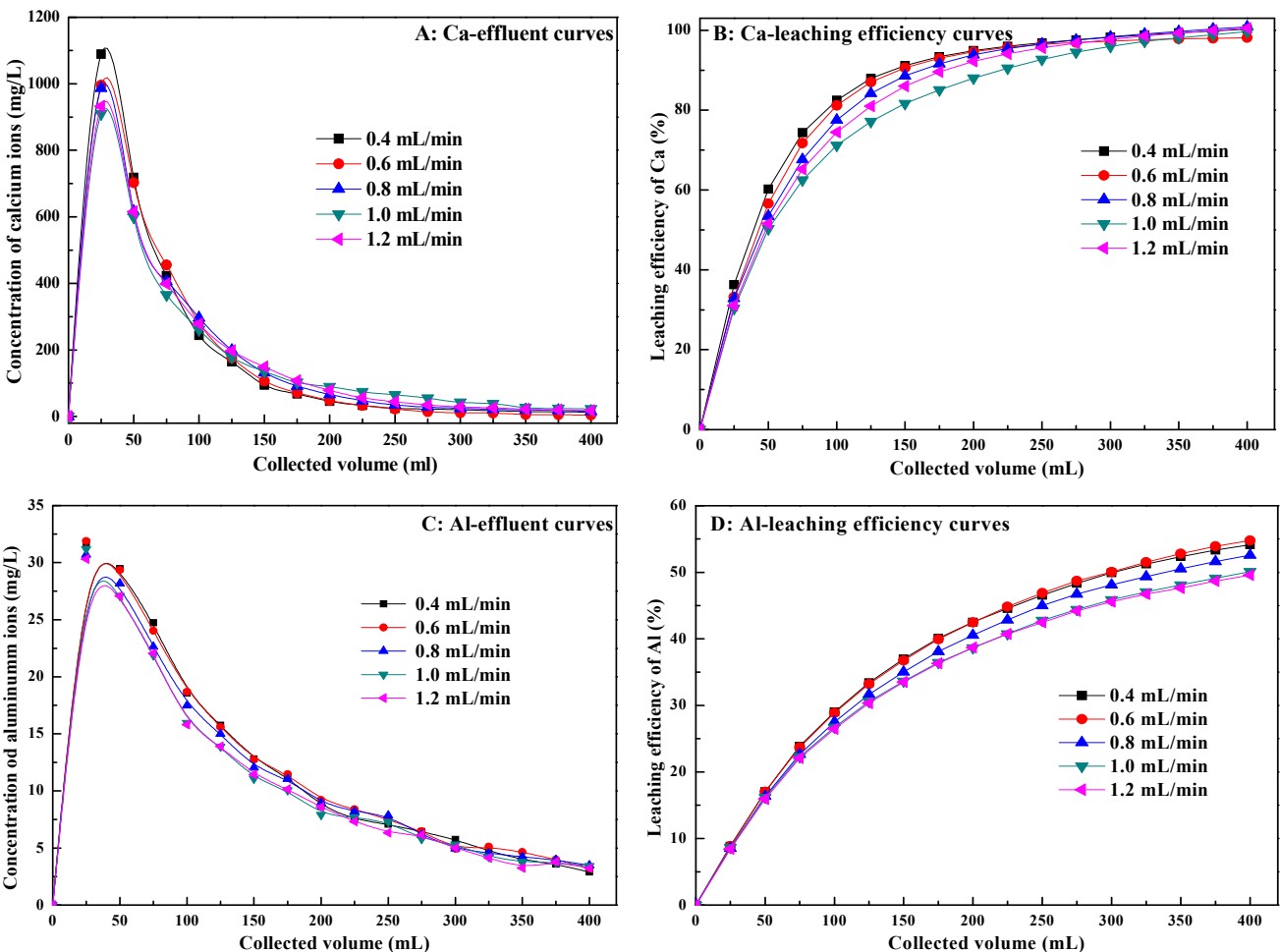

**Figure 12.** Column leaching behaviors of calcium and aluminum with different flow rates of the leaching agent (T = 20 °C, $C_0$ = 0.20 mol/L, pH = 5.70). (**A**) Ca-effluent curves, (**B**) Ca-leaching efficiency curves, (**C**) Al-effluent curves, (**D**) Al-leaching efficiency curves).

### 3.3.3. Effect of Leaching Temperature on the Leaching Behaviors of Calcium and Aluminum

The effect of the leaching temperature on the leaching behaviors of calcium and aluminum was developed. The IEP-Ca and IEP-Al are adsorbed on clay minerals via electrostatic action, generally a physical adsorption process, and are rapidly desorbed in an $MgSO_4$ leaching agent at a concentration of 0.2 mol/L. Therefore, increasing the leaching temperature has a limited effect on the desorption of IEP-Ca and IEP-Al, which can be observed from the data of the apparent reaction rate constants and activation energy in the leaching kinetics, shown in Figures 6 and 9.

The leaching of $Ca^{2+}$ was only affected by the desorption reaction that emerged as an endothermic process controlled by internal diffusion. Therefore, the migration speed of $Mg^{2+}$ and $Ca^{2+}$ increased with the leaching temperature. Meanwhile, the reaction in Equation (1) shifted to the right, which slightly increased the peak concentration of $Ca^{2+}$ in the leachate and the leaching rate of $Ca^{2+}$, as shown in Figure 13A,B. In addition, with the increase in leaching temperature, the collected volume of the leachate required for the leaching efficiency of calcium to reach 98% also decreased.

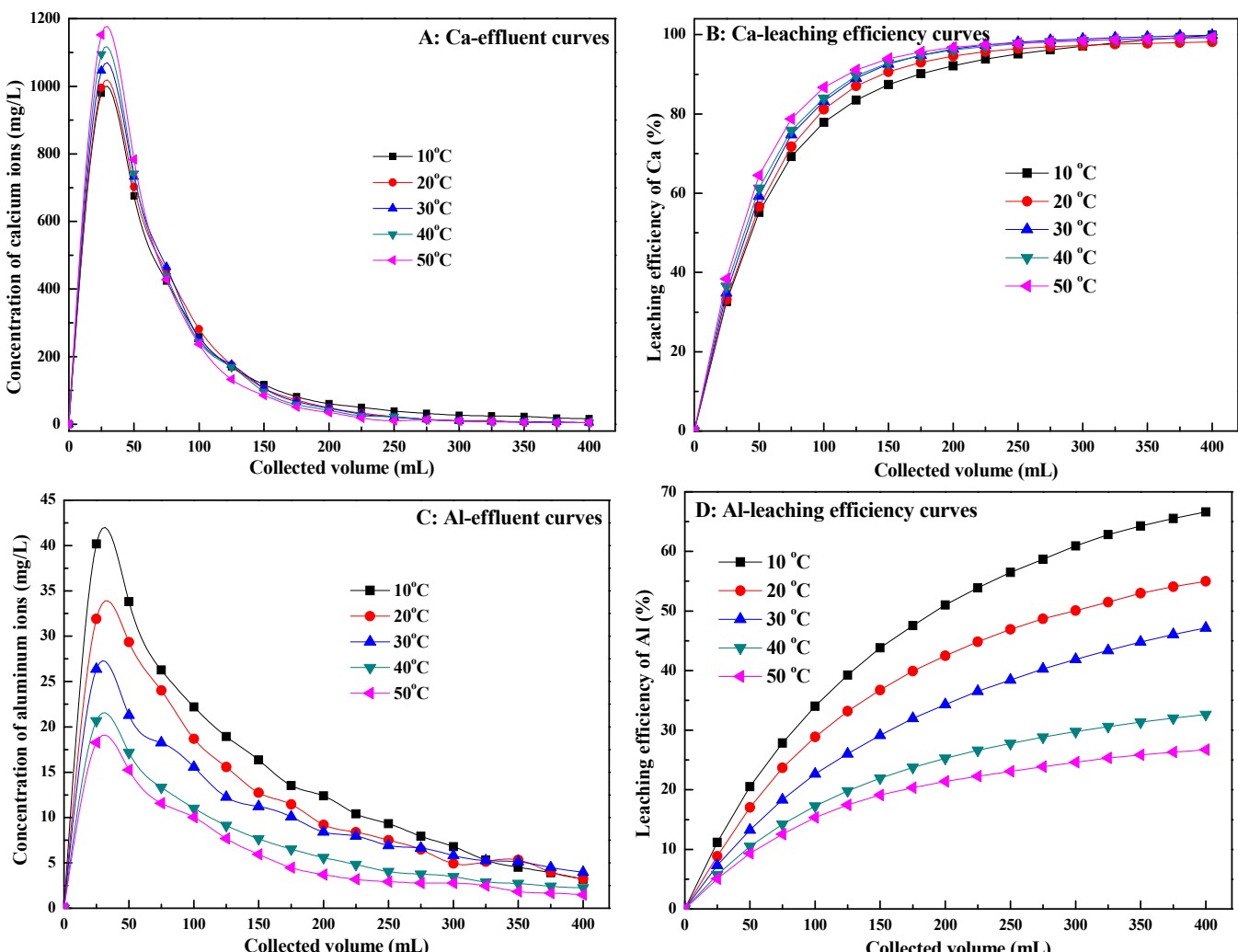

**Figure 13.** Column leaching behaviors of calcium and aluminum at different leaching temperatures ($C_0$ = 0.20 mol/L, $\upsilon$ = 0.60 mL/min, pH = 5.70). (**A**) Ca-effluent curves, (**B**) Ca-leaching efficiency curves, (**C**) Al-effluent curves, (**D**) Al-leaching efficiency curves).

Relatively, the hydrolysis reaction of $Al^{3+}$ is accompanied by ion exchange in the leaching process. On one hand, when the concentration of $MgSO_4$ was 0.20 mol/L, the leaching temperature had a limited effect on the ion exchange of IEP-Al. Therefore, as per Figure 13C, the volume-to-peak of aluminum at different temperatures is close. On the other hand, the hydrolysis of $Al^{3+}$ is an endothermic process. When the temperature increased, the hydrolysis equilibrium (Equations (7)–(12)) shifted to the right and more aluminum was in the liquor form of aluminum hydroxide [32,33] and remained in the raw ore. Therefore, Figure 13C shows that the peak concentration of aluminum in the effluent curves is smaller at a higher temperature, which is consistent with the phenomenon observed in the previous leaching process by ammonium sulfate [15]. Simultaneously, Figure 13D shows that the leaching efficiency of aluminum gradually decreased with the increase in leaching temperature. For example, when the collected volume of the leachate was 400 mL, the leaching efficiency of aluminum was reduced from more than 65% to about 25% when the leaching temperature rose from 10 °C to 50 °C. There is an obviously good consistency in the leaching kinetics of aluminum.

Therefore, because the IEP-Ca is more easily desorbed than rare earth, and there is no other reaction in the leaching process, calcium is inevitably leached under different temperature conditions. For $Al^{3+}$, the hydrolysis reaction is intensified by increasing the leaching temperature, so a lower leaching efficiency of aluminum is obtained at a

higher leaching temperature. For example, it may can be a good choice to leach ICREO in the summer.

**4. Conclusions**

(1) The leaching process of calcium from ICREO with $MgSO_4$ is a physical ion-exchange process of IEP-Ca, which is controlled by internal diffusion. The activation energy is 8.97 kJ/mol. With the increase in the $MgSO_4$ concentration, the leaching rate of calcium increases, so the volume-to-peak of the calcium concentration in the leachate will shift to an earlier time, and the peak concentration and leaching efficiency of calcium will increase. However, the flow rate and leaching temperature have a limited effect on calcium leaching. With the decrease in the flow rate and increase in temperature, the peak concentration of calcium slightly increases.

(2) There are two reactions in the leaching process of aluminum with $MgSO_4$: ion exchange and hydrolysis. The leaching reaction is controlled by internal diffusion, and the activation energy is 10.48 kJ/mol. The hydrolysis reaction of $Al^{3+}$ is greatly affected by the concentration of $Al^{3+}$ and reaction temperature. Raising the $MgSO_4$ concentration and leaching temperature will intensify the hydrolysis reaction, which rapidly decreases the leaching efficiency of aluminum.

(3) Because calcium has a faster leaching rate than rare earth, IEP-Ca will inevitably be leached into the leachate when most of the rare earth is leached by $MgSO_4$. For aluminum, its hydrolytic tendency can be improved by increasing the leaching agent concentration and leaching temperature, so it can effectively reduce the content of aluminum in the leachate.

**Author Contributions:** Conceptualization, Q.H. and Y.X.; data curation, Q.H. and J.Q.; formal analysis, Q.H. and J.Q.; funding acquisition, Y.X.; investigation, Q.H., J.Q., and M.R.; methodology, Y.X.; project administration, Y.X.; resources, Y.X.; supervision, M.R.; validation, J.Q.; visualization, Q.H. and M.R.; writing—original draft, Q.H. and Y.X.; writing—review and editing, Q.H. and Y.X. All authors have read and agreed to the published version of the manuscript.

**Funding:** The authors gratefully acknowledge the financial support of the National Key Research and Development Project of China (2019YFC0605002, 2020YFC1909002), the National Natural Science Foundation of China (51604128), Jiangxi Province's "double thousand plan" (jxsq2019201116), the Youth Jinggang Scholars Program in Jiangxi Province (QNJG2019056), the Key R&D Programs of Science and Technology Project of Ganzhou City ([2017]179), the Science and Technology Innovation Talents Program of Ganzhou City ([2018]50), and the cultivation project of the State Key Laboratory of Green Development and High-Value Utilization of Ionic Rare Earth Resources in Jiangxi Province (20194AFD44003).

**Acknowledgments:** The author thanks the members of the Institute of Green Metallurgy and Process intensification (GMPI) for their selfless help.

**Conflicts of Interest:** The authors declare no conflict of interest.

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
