# Peer review of "Leaching Behaviors of Calcium and Aluminum from an Ionic Type Rare Earth Ore Using MgSO4 as Leaching Agent"

_minerals, doi:10.3390/min11070716_

Round 1

Reviewer 1 Report

This paper addresses the ion-exchange mechanism of various metal ions including rare earth elements (REEs) from a certain type of REE-bearing ores. It reads well and experimental procedure is commendable, and the analysis of the results obtained is sound and convincing. The reviewer recommends the paper be accepted for publication in current form.

Author Response

Thank you very much for your positive and encouraging comment. Once again, we have further checked the grammar.

Reviewer 2 Report

The article is devoted to an interesting topic on the study of the leaching of impurities from a new type of raw material - ionic type rare earth ore. The paper provides new data on the behaviour of calcium and aluminium in the leaching process, which will undoubtedly be of interest to readers, but the article has a number of drawbacks:

l. 12 The equation should go immediately after the reference to it in the text;

l. 52 What on the other hand?

l. 64 If the analysis was carried out using ICP, it would be more logical to present the data in ppm.

l. 66 Maybe with a low Eu content?

l. 67 You should specify the X-ray apparatus and the databases.

l. 72. The peaks of montmorillonite and kaolinite coincide, it is necessary to specify on the basis of which both are selected. What are the phases at the angles 25, 40, 45, 55o? These peaks are quite large and should also be deciphered.

l. 81 Heated water will give 10 oC? You should specify how this temperature was obtained in the experiments.

l. 112 "Calcium sulfate precipitate...."

130 You should also use "precipitate".

l. 152 Precipitation occur, the precipitate is formed.

l.170. As you repeatedly point out, the process consists of the desorption of ions from the surface of aluminosilicate minerals, i.e. the aluminosilicate matrix itself is not attacked during leaching and there is no shrinkage of the core, so the model of the shrinking core, in this case, can not be used. Accordingly, the conclusions drawn further are based on the application of these equations.

l. 180 This equation is questionable.

l. 319 The proposal must be rewritten

See the attachments for an understanding of the position of the lines.

Author Response

  Thank you for your precious comments concerning our manuscript. Those comments are very helpful for revising and improving our paper. We have studied comments carefully and have made corrections which we hope meet with your approval. Revised portions are marked in red in the paper. The responses to the reviewer’s comments are given in the attached file.

Round 2

Reviewer 2 Report

The authors have corrected all comments, the arguments given are convincing, so I recommend the article for publication.

This manuscript is a resubmission of an earlier submission. The following is a list of the peer review reports and author responses from that submission.

Round 1

Reviewer 1 Report

The ion-exchange mechanism is relevant to the extraction of rare earth elements (REEs) from some of REE-containing ores. This paper addresses this aspect which is commendable. However, the paper has serious flaws that should be addressed before it can be considered for publication.

The title of the paper is not clear and should be changed. For example, “Leaching behavior of Ca and Al from an ionic type of rare earth ore using MgSO4 as an ion-exchanger.”

Introduction: This study is an extension of earlier studies, in which the following conclusions might have been arrived. About 80% of REEs are currently being leached using Na+, NH4+ and Mg2+ by an ion-exchange mechanism. The current investigation is addressing the leaching of Ca2+ and Al3+ using Mg2+ as the exchange reagent at pH of about 4.5-5.0, which should be proceeded the leaching of REEs.

If the above assumptions are correct, it is unclear how this study has an impact on the extraction of REEs. It appears to the potential readers that the study may be an academic exercise but its impact, in practical implication, in regard to the extraction of REEs is unclear and it could be uneconomical and impractical. This aspect should be explained.

Section 3-2: The authors described that the kinetics of the ion-exchange process is governed by a shrinking core model. Simply because experimental data fit in a model in the literature, the system belongs to the same mechanism is one of the worst assumptions one can make. The authors should study the origin description of the “Shrinking Core Model” and explain the assumptions involved in the model are applicable for their system. This reviewer can hardly see any justification be made on any adsorption system following the Shrinking Core Model. Adsorption process especially when ion exchange mechanism plays a role would more likely be a diffusion process. Although the shrinking core model does include diffusion aspects as a part of the process, the circumstance is quite different. If the authors believe otherwise, they should justify why that is. This section and the following sections related to this assumption should be re-evaluated and rewritten.

Line 494: The statement, “the IEP-Ca is more easily desorbed than rare earth” should be qualified. Please explain on what ground this statement us valid.

English should be checked carefully by a person whose first language is English. Some paragraphs are too long for the readers to comprehend.

Use % for percentage and not the other symbol used (line 110) in this text.

Reviewer 2 Report

The manuscript reported results about “Behavior of impurities in the leaching process of the ionic type rare earth ore by MgSO4” The paper is interesting and easy to follow. Data are consistent with the main conclusions of the work. As it is known that calcium and aluminium lead to an increase in precipitant agent consumption and rare earth loss. The present work indicates that calcium is taken into the leach solution along with rare earth elements due to the faster leaching rate but aluminium can be rejected from the leach solution by increasing the leaching agent concentration and leaching temperature during the leaching process. From my point of view, the manuscript should be appreciated for publication. It would be of interest to the Journal.